# Neural Classifier of Deep Brain Stimulation Evoked Emotion

## Abstract

Precise deep brain stimulation (DBS) of Subcallosal Cingulate White Matter (SC-Cwm) alleviates symptoms of treatment resistant depression (TRD). Objective signatures from neural recordings are needed to optimize implantation and programming of antidepressant brain stimulation, and recent advances in machine learning help identify these in noisy patient recordings. In this study, we present a machine learning classifier build from previously reported dense-array scalp EEG taken during active DBS at therapeutic (OnTarget) and non-therapeutic (OffTarget) targets. Using combined emotion self-report and EEG measurements alongside OnTarget stimulation of SCCwm and OffTarget stimulation $1.5\,\mathrm{mm}$ away, we trained a *support vector machine* (SVM) capable of confirming precise stimulation. We demonstrate that the learned model coefficients align with *engaged tractography* predicted through volume of tissue activated (VTA) modeling. This compound model will enable implementation, study, and improvement of adaptive SCCwm-DBS, particularly in TRD, more systematic. The classifier is released open source to the community for further validation, refinement, and extension; the dataset is released as a part of a multimodal foundation for antidepressant DBS.

## 1 Introduction

Deep brain stimulation (DBS) has emerged as a promising, albeit complex, therapeutic option for severe, treatment-resistant psychiatric depression (MDD) in a subset of patients who have not responded to conventional therapies Mayberg et al. (2005); Riva-Posse et al. (2014; 2018). While the clinical efficacy of DBS for MDD has been demonstrated in open-label studies and meta-analyses, a number of large randomized controlled trials have yielded mixed outcomes, highlighting the need for more systematic and precise methods to guide and optimize therapeutic targeting Holtzheimer et al. (2012); Mayberg et al. (2005); Riva-Posse et al. (2014; 2018); Howell et al. (2019).

Systematic study and validation of antidepressant DBS within the SCC requires more objective signatures specific to therapeutic efficacy Smart et al. (2018); Tiruvadi et al. (2022a); Waters et al. (2018); Starr (2018); Stanslaski et al. (2012; 2018). Since an equivocal trial, evidence has emerged that broad DBS of the SCC is insufficient to achieve sustained antidepressant response - rather precise stimulation of SCCwm is needed, likely with individualized tractography. The subcallosal cingulate white matter (SCCwm) has been a primary target for DBS in MDD, based on its role as a critical node within a mood-regulating brain network. However, the mechanism by which stimulation of this region alleviates symptoms is still an active area of investigation, and the precise anatomical and electrophysiological signature of effective stimulation remains unclear. Early investigations have identified a network of brain regions are likely targets for successful antidepressant DBS Tiruvadi et al. (2022a;b); Waters et al. (2018) but data heterogeneity and the high dimensional variances of clinical care make standard analyses insufficient.

In this study, we present a machine learning-based approach to address these challenges by leveraging scalp neural recordings to identify and validate a precise stimulation target. We introduce a regularized support vector machine (SVM) classifier capable of distinguishing precise DBS of the SCCwm ("OnTarget") from nearby stimulation ("OffTarget"), as confirmed by emotional self-responses and VTA analysis. This work advances the field of neuromodulation by providing a

foundational model that can serve as an anchor for future systematic clinical trials and rational engineering of adaptive DBS systems.

## 2 METHODS

### 2.1 PATIENTS AND OUTCOME

Six consecutive patients were implanted between June 1, 2013 and January 1, 2017 as a part of an IRB approved research protocol at Emory University studying the SCCwm-DBS for TRD (ClinicalTrials.gov Identifier NCT01984710) using inclusion and exclusion criteria. Written informed consent was provided by each patient to participate in the study protocol (FDA IDE G130107) and the study was continuously monitored by the Emory University Department of Psychiatry and Behavioral Sciences Data and Safety Monitoring Board. All patients completed the PC+S™ study, continuing until the battery was depleted, and were treatment responders beyond 12 months Stanslaski.

### 2.2 TRACTOGRAPHY AND IMPLANTATION

Pre-operative diffusion tractography was used to individually target DBS implants to a specific intersection of four white matter bundles in the subcallosal cingulate white matter (SCCwm). During implantation, each bilateral DBS lead was placed to ensure an electrode precisely in the target ("OnTarget") for therapy, leaving another adjacent ("OffTarget") as an experimental contrast $1.5\,\text{mm}$. The neural structures activated by each electrode were then computationally modeled using an "engaged tractography" technique, which analyzes the volume of tissue activation in relation to the patient's unique white matter anatomy.

### 2.3 DBS AND EXPERIMENT

Patients were brought in four weeks after implantation, with chronic stimulation inactive throughout, for an experiment day and subsequent therapeutic stimulation initiation. Chronic, therapeutic stimulation was initiated at $3.5\,\text{V}$ and $130\,\text{Hz}$, $90\,\mu\text{s}$ pulsewidth, monopolar at the OnTarget stimulation site *bilaterally*. Experimental stimulation was performed at both OnTarget and OffTarget targets at $6\,\text{mA}$; otherwise identical parameters as chronic, therapeutic stimulation. Stimulation was delivered for $3\,\text{min}$ with $1\,\text{min}$ washout immediately before and after, and a $20\,\text{min}$ washout was done between Targeting changes.

### 2.4 EMOTION SELF REPORT

Patients were blinded to stimulation condition (On or Off) and configuration (OnTarget, OffTarget, laterality). Patients were presented with three buttons and told to press the button best corresponding to their emotion and/or sensation at any time. Button presses were summed within each stimulation condition and reported as associated with the corresponding configuration.

### 2.5 NEURAL RECORDINGS

EEG Acquisition Dense-array electroencephalography (dEEG) timeseries were collected with a 256-channel Hydrocel Geodesic Sensor Net (Electrical Geodesics Inc., Eugene, OR) and Netstation data-acquisition software (Electrical Geodesics Inc., Eugene, OR). Channels are sampled at 1 kHz and referenced against Cz. All electrode impedances were maintained below $50\,\text{k}\Omega$ and checked every 20 min. Hardware filtering was performed at 100 Hz low-pass and a 0.1 Hz highpass to remove just high-frequency noise and baseline DC offsets. Patients were seated comfortably in a climate-controlled environment with head positioned with an adjustable chin rest. During all recordings patients were told to blink naturally, and relax muscles in shoulders, neck, face.

### 2.6 OSCILLATORY POWER FEATURES

**Time-Frequency and Oscillatory States** All recordings are transformed into the time-frequency domain using STFT and overlapping Welch windows. Parameters were kept constant for the Welch

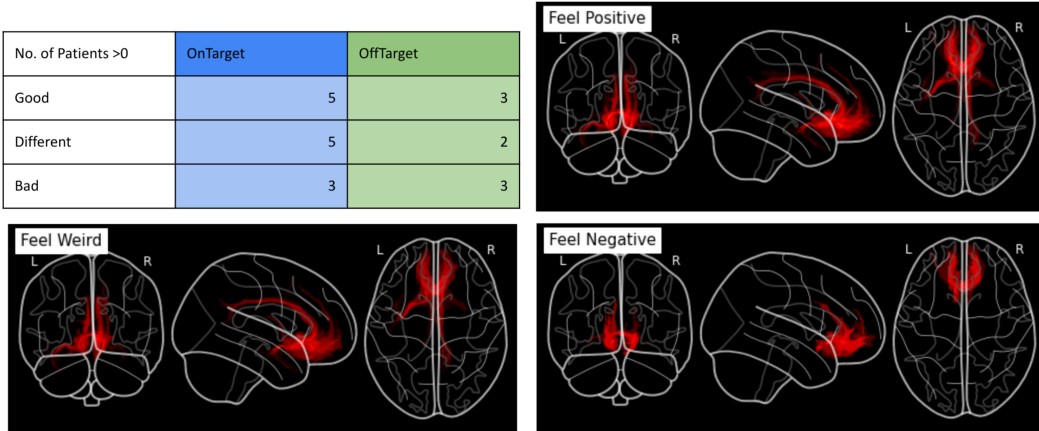

Figure 1: **Self-reported Emotion Changes**. Top Left: Table of responses under OnTarget and Off-Target stimulation. OnTarget exhibited more overall responses than OffTarget, and these responses were mostly not 'bad'. Others: The engaged tractography associated with each of the self-reported emotions evoked.

estimate of the PSD, and oscillation frequency windows are fixed at the MC adjusted ranges - consistently across both LFP and EEG. We calculate these features for all LFP+EEG channels (258 total) across all segments (total of approximately 600, depending on stringency of preprocesing). The brain's instantaneous oscillatory state is measured as a vector $\vec{\theta}_t \in \mathbb{R}^{256 \times 4}$.

**Oscillatory Responses**  Median power was calculated in each oscillatory feature across all pre-stimulation segments Tiruvadi et al. (2022b) This median power is subtracted from each peri-DBS segment, yielding a *response vector*. Response vectors are used for classifier training and assessment, and all segments from the EEG patient subcohort are used to build a training curve and determine the optimal number of segments to use in classifier training.

## 2.7 SUPPORT VECTOR MACHINE

A support vector machine (SVM) was chosen for its robustness in classifying high-dimensional brain signals. We implemented a 2-class linear SVM with an $L_1$ regularizer using Python's `scikit-learn` library to map baseline-corrected oscillatory responses from EEG channels to their corresponding stimulation class, either {OnTarget} or {OffTarget}. The classifier was trained on a balanced set of data segments and tested on subsampled segments from a held-out testing set. The use of an $L_1$ regularizer also enabled the identification of a sparse subset of EEG channels for the classification task.

## 2.8 CODE AVAILABILITY

Code is openly available at **redacted**. Model coefficients and associated architecture are available at **redacted**

## 3 RESULTS

### 3.1 EMOTIONS EVOKED

Patient self-reported button presses demonstrate a clearly differential pattern across stimulation target (Figure 1). OnTarget is predominantly not 'bad' while OffTarget is evenly split. The response rates are too small to perform robust statistical testing, but speculation from early trends may align with short-term electrophysiology Sendi et al. (2021). In all cases, patients are placed OnTarget chronically - n=5/6 patients were long-term responders to treatment at 6 months, all n=6/6 patients were responders at 12+ months.

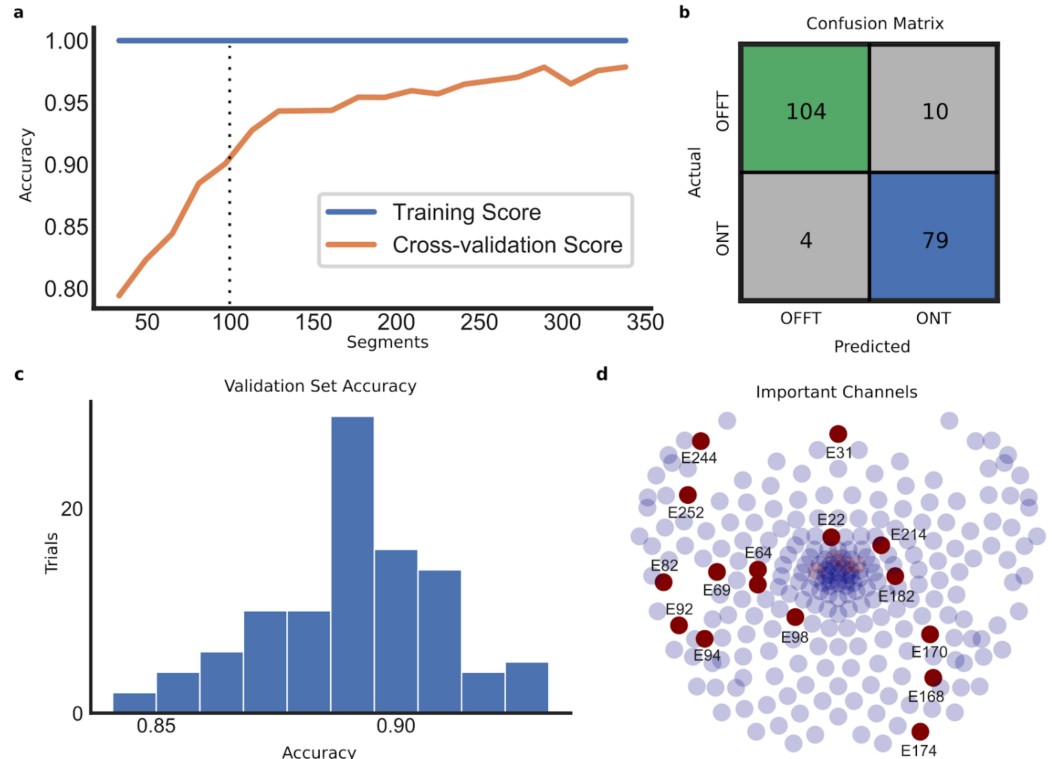

Figure 2: **Classifier Training, Error, and Performance**. a) Training curve demonstrates quick convergence to 90% accuracy with only 30% of segments. b) Errors are predominantly misclassification of OffTarget as OnTarget. c) Bootstrapped validation set across 100 trials demonstrates high accuracy in unseen segments. d) L1-regularized SVM coefficient magnitudes across all oscillatory powers align along left temporal and right parietal dEEG channels.

## 3.2 TARGETING CLASSIFIER

**Training**   The trained $L1$-regularized SVM achieved a high performance inside the training set, with performance plateauing at 0.95 with 33% of the available segments used to train (Figure 2a). The final SVM is trained on 100 random segments (Figure 2a).

**Testing and Performance**   In a single assessment trial consisting of the remaining 250 segments, the classifier achieves 90% accuracy in binary classification (Figure 2b). With a bootstrapped estimate of the classifier accuracy, an average of 0.89 accuracy in binary classification is found over 100 trials (2c).

**Errors and Coefficients**   Errors are primarily in misclassification of OffTarget as OnTarget (Figure 2b) and not vice-versa. Overlap in tractography may explain this as OnTarget and OffTarget are likely to hit some shared tracts (Figure 3). The coefficients lie largely along the left temporal and right parietal regions (Figure 2d).

## 3.3 TRACTOGRAPHIC ALIGNMENT

**Useful Channels**   To isolate the most informative EEG channels the SVM training procedure is regularized with an $L_1$ penalty (Figure 2d). The resulting mask demonstrates an asymmetry: left temporal channels and right parietal channels are in the top 10% of largest coefficient $L_2$ norms.

**Whole-brain Connectomic Masking**   We propose a final filtering step to yield a *direct antidepression circuit* mapped from the direct effects of antidepressant OnTarget SCCwm-DBS (Figure

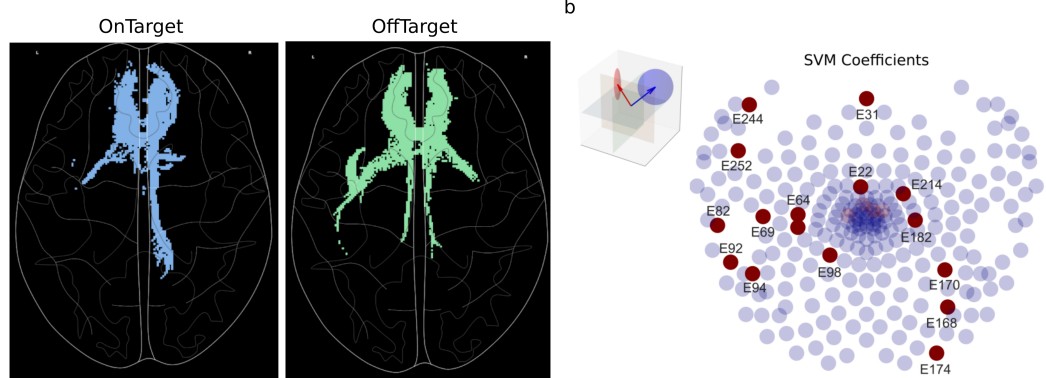

Figure 3: **Alignment of Targeted Tractography and SVM Coefficients.** a) OnTarget and OffTarget Engaged Tractography difference. Demonstrates which voxels are engaged more under OnTarget (blue) and OffTarget (green), respectively. b) Top 10% of classifier coefficients in a max across any oscillatory band calculation. Follow a left temporal and right parietal asymmetric pattern.

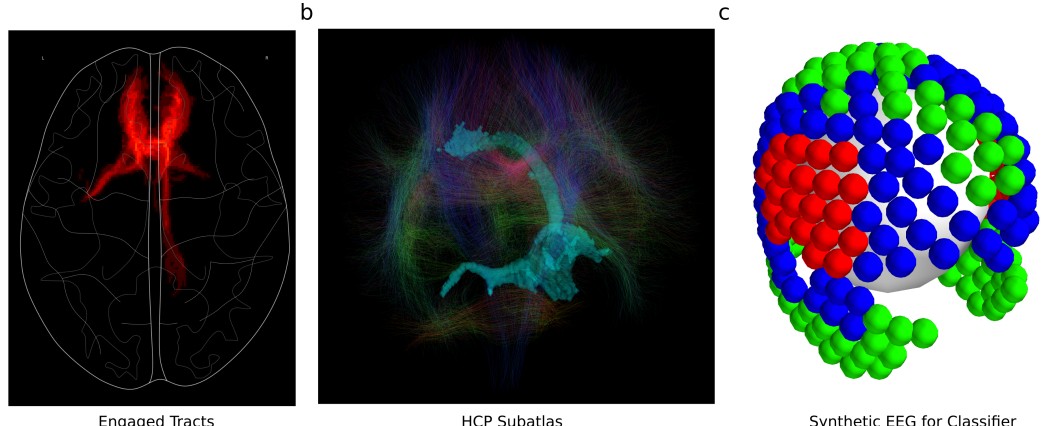

Figure 4: **Pipeline for Generative Connectomics.** a) Engaged tracts under OnTarget stimulation. b) Engaged tractography mask is used to filter the Human Connectome Project (HCP900) structural connectome. c) Filtered tracts can be used to generative synthetic EEG signals for comparison with empirical oscillatory changes.

4). To do this, we coregistered our engaged tractography map, with binary threshold, with an openly available HCP-based dMRI connectome Li et al. (2020) to identify the key fibers involved (Figure 4b). Masks for OnTarget and OffTarget are used to filter out the streamlines associated with each of the emotions evoked in Figure 1 Table. The resulting synthetic EEG can be used to fine tune the SVM based on patient-specific dMRI (Figure 4c).

## 4 DISCUSSION

Classifiers that can use neural recordings to confirm adequate DBS implantation and stimulation are needed for more systematic implementation and study of therapy, especially in psychiatric disorders like depression. In this study, we reanalyzed a previously reported set of data Tiruvadi et al. (2022b) using ML to build a classifier capable of determining precise SCCwm-DBS, and then linking the classifier coefficients to tractography and self-reported emotion.

Noninvasive recordings provide evidence that stimulation just $1.5\,\mathrm{mm}$ away from the target can evoke a distinctly different neural state, measured in both temporal evoked potential Tiruvadi et al. (2022a) and spectral responses Tiruvadi et al. (2022b). Our classifier provides an alternative look at

that evidence, with a more direct link between engaged tractography and scalp response (Figure 3). By interpreting electrophysiology through the lens of anatomy, we use the tractography from surgical planning as a scaffold to analyze brain-wide oscillatory responses to SCCwm-DBS. Calculation of engaged tractography confirms that OnTarget stimulation engages asymmetric white matter tracts, specifically the right cingulum bundle (right-CB), which emerges as a major differentiator consistent with a growing body of literature Howell et al. (2019); Riva-Posse et al. (2018); Tsolaki et al. (2021). In contrast, OffTarget stimulation engages more symmetric tracts and the left uncinate fasciculus (left-UF). The observed $\alpha$ response and other oscillatory dynamics align spatially with these engaged tracts, suggesting that SCCwm-DBS evokes dynamics reflecting the structural and synaptic processes of the underlying networks, which is consistent with a biphasic mechanism of action and has implications for linking brain structure to changes in mood.

Leveraging these findings, we developed a supervised learning model as a potential target engagement classifier to distinguish neural signals caused by therapeutic OnTarget versus nearby OffTarget stimulation. A linear classifier achieved approximately 90% testing accuracy in this task (Figure 2) and identified right-parietal and left-temporal channels as the most informative, corroborating the tract differences found in our DTI analysis (Figure 3).

This study has several limitations. First, the small sample size of six patients is limiting to generalization. However, the full population of patients that receive SCCwm-DBS is small, and this is a study of a sizable portion of that population. Second, The low number of self-report emotion button clicks limits our ability to make strong statements about the exact nature of OnTarget and OffTarget effects on emotion. Third, the SVM performance is reported in-sample due to limited size. Additionally, while training and testing never overall in the temporal segment used, they are still taken from the same task and intrinsic autocorrelations could bias results. The alignment of coefficients to tractography is taken as a mitigation of this, but the limitation must be addressed moving forward. Finally, it remains unclear if engagement of OnTarget at $6\,\mathrm{mA}$ is desired as this is above therapeutic values. Attempting to evoke the same state as suprathreshold stimulation may not be therapeutic, and a more direct experiment at therapeutic parameters will be done.

## 5 CONCLUSION

We trained an ML classifier on previously reported scalp EEG signatures of precise SCCwm-DBS to build a model capable of identifying OnTarget SCCwm-DBS from scalp electrodes. While further testing is needed, the classifier's performance and its alignment with anatomical tractography provide preliminary confidence in this approach for developing a physiologic readout for adaptive DBS therapy in MDD and/or SCCwm-DBS. Future work will focus on optimizing classification while minimizing the number of EEG electrodes, and more explicitly integrating neural dynamics into the classification task Tiruvadi et al. (2022a), potentially as ANNs or latent state space models. The classifier will be released open source to the community for further validation, refinement, and extension to enable more reliably implantation and engineer adaptive antidepressant DBS systems.

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

## A Appendix

