# OpenReview forum: "Neural Classifier of Deep Brain Stimulation Evoked Emotion"
_ICLR.cc/2026/Conference — Submitted to ICLR 2026_

### Official Review · Reviewer_bdZh · 2025-10-28

**Soundness:** 1
**Presentation:** 1
**Contribution:** 2
**Rating:** 0
**Confidence:** 4

**Summary:**

The article proposes a method to determine an effective deep brain stimulation treatement using high density scalp EEG.  For an effective treatment stimulation of the white matter in a particular part of the brain (subcallossal cingulate white matter or SCCwm) is needed.  A short-term fourier transform of scalp EEG is calculated and supplied to a support vector machine algorithm to determine if the deep brain stimulation occurred in the SCCwm.

The main contribution is the presentation of the SVM algorithm to distinguish an "OnTarget" and "OffTarget" stimulation.  The authors have tested their algorithm in real data, although the sample size is small.  This type of dataset is very unique.

**Strengths:**

- The paper presents a relevant problem within the efficacity of deep brain stimulation.  The originalty lies in the fact that the authors used a machine learning algorithm to perform this specific task of distinguishing two types of stimulation.

**Weaknesses:**

The article is very concise and a lot of aspects of the methodologies are not explained. Such as:
- It is not clear how the SVM is trained or what the gold standard data is.  The authors should be more elaborate on how the SVM was trained.
- The feature construction from the data is not clear.  The paper lacks a thorough description on how the raw data was transformed into the feature space.  Details about the preprocessing is important to understand the results.  To me it is not clear how they come from a 258 signals to a feature set of $\theta_t \in \mathbb{R}^{256\times 4}$.  The authors should explain step by step how the features are calculated, providing all details on the Short Term Fourier Transform.
- it is also not clear how the cross validation was done.
- The sample size is very small.  the authors discuss a possibility to extend the data with synthetic EEG, but no results are shown.  This is an interesting aspect which has potential.
- Many abbrevations are not explained.
- The article lacks a thorough experimental design, execution and validation.

**Questions:**

- What do the authors mean by oscilatory state?   How was this state obtained?
- How is the cost function look like?  What terms are in the L1 regularisation.  Regularisation using the L1 is done usually to pervent overfitting, but will it be sufficient to cope with the overfitting due to small sample size?

---

### Official Review · Reviewer_gg3n · 2025-10-29

**Soundness:** 3
**Presentation:** 3
**Contribution:** 2
**Rating:** 4
**Confidence:** 4

**Summary:**

This paper presents a machine learning framework for classifying deep brain stimulation (DBS) targets based on dense-array EEG recordings in patients with treatment-resistant depression (TRD). The authors train a regularized support vector machine (SVM) to distinguish between OnTarget (therapeutic subcallosal cingulate white matter stimulation) and OffTarget (1.5 mm away) conditions. The classifier achieves approximately 90% accuracy and aligns with engaged tractography predicted through volume of tissue activated (VTA) modeling, providing a physiologically interpretable readout for DBS targeting.

**Strengths:**

1. The primary strength is the innovative use of a regularized SVM on scalp EEG to non-invasively classify the precise, therapeutic DBS target (OnTarget vs. OffTarget).
2. The methodology is clear and replicable, including detailed signal processing (STFT, Welch windowing), model specification (L1-regularized SVM), and data acquisition parameters.
3. Furthermore, the authors commit to an open-source release of the code and dataset, which promotes reproducibility and future extension.

**Weaknesses:**

1.	The primary weakness is the small sample size ($n=6$ patients). While acknowledged, this heavily limits the confidence in the model's performance on a broader patient population.
2.	Since training and testing segments were derived from the same experimental sessions, the classifier may be at risk of overfitting due to temporal autocorrelations, and its true generalization ability across subjects remains uncertain.
3.	The classifier was trained using data from suprathreshold stimulation (6 mA), which is higher than the therapeutic chronic stimulation voltage (3.5 V). This raises the question of whether the neural signature learned is truly the one associated with the therapeutic effect, or simply a maximal effect. This limits the immediate clinical applicability.
4.	Lack of cross-validation or independent testing. The classifier’s evaluation is based on intra-session data splits rather than subject-level or session-level cross-validation. Without testing on unseen subjects or external datasets, it is difficult to assess the model’s robustness to inter-patient variability.

**Questions:**

1. The paper mentions that performance is reported "in-sample due to limited size" and that a held-out testing set of the remaining 250 segments was used after training on 100 random segments. Given the concern about bias from autocorrelations, did the authors attempt a Leave-One-Patient-Out Cross-Validation (LOPOCV)? If so, what was the average accuracy? If not, could this be explored as a more rigorous test of generalizability across individuals?
2. How significant is the classifier’s performance relative to chance, given the small sample size? Have the authors performed permutation testing to estimate p-values or confidence intervals?
3. Could the authors provide frequency-band–specific coefficient maps to show whether certain oscillatory bands (e.g., $\alpha$, $\beta$ ) dominate discrimination?
4. The authors note that engagement at 6 mA is above therapeutic values and that a more direct experiment at therapeutic parameters will be done. Can the authors discuss, or speculate, how the classifier's performance and coefficient map might change if the training data were collected at the chronic therapeutic parameter of 3.5 V/130 Hz/90 $\mu$s (or equivalent current)?
5. The authors should provide more experimental results and analyses to enhance the validity and robustness of the conclusions. This should include more evaluation metrics, comparative experiments with non-regularized linear models, feature ablation studies (such as only using time-domain features), and so on.

---

### Official Review · Reviewer_PbVB · 2025-10-31

**Soundness:** 1
**Presentation:** 3
**Contribution:** 2
**Rating:** 2
**Confidence:** 4

**Summary:**

The paper studies whether scalp EEG can distinguish precise (“on-target”) vs nearby subcallosal cingulate DBS stimulation. A sparse linear SVM is trained on band-power features and shows strong within-session discrimination. The authors qualitatively relate learned spatial patterns to tractography and outline a future connectomics-driven augmentation idea, with plans to release code/data.

**Strengths:**

1. Important clinical problem (target engagement for SCCwm-DBS) and a carefully conceived on- vs off-target manipulation.

2. Nontrivial data collection under challenging conditions; protocol looks promising for future work.

3. Transparent model choice (linear, sparse) that encourages interpretability and potential reproducibility.

4. Initial anatomy-physiology linkage (EEG patterns vs tractography) that could become meaningful with stronger evidence.

**Weaknesses:**

1. Reliance on random within-session splits risks temporal/identity leakage; there is no leave-one-subject/session-out evaluation to demonstrate generalization. The 100% train accuracy suggests potential overfitting in case of which such leakage can “hide” overfit and overinflate testing metrics

2. Only one classifier is explored; there are no simple linear baselines, permutation/time-shift controls, or subject-ID/recording-day checks.

3. Interpretability is mostly qualitative. No weight-stability, ablation, or quantitative alignment analyses are provided.

4. Very small N with window-level inference rather than subject/session-level statistics; claims are stronger than the evidence supports.

5. “Foundation model” positioning is not warranted by scope or results; the connectomics augmentation remains conceptual.

6. Under-specified artifact handling. Concurrent DBS-EEG can introduce stimulation-locked and EMG artifacts; the mitigation pipeline isn’t detailed enough to rule out non-neural contributions.

7. Reproducibility unclear. Code/data/weights are not yet available for verification.

**Questions:**

1. Can you report leave-one-subject and/or leave-one-session/day-out results with subject-level statistics to establish leakage-safe generalization?

2. Do results persist under simple linear baselines (e.g., logistic/elastic-net, ridge) and sanity checks (label permutation, time-shift, subject/recording controls)?

3. What is the complete artifact-mitigation pipeline during DBS-EEG (filtering, referencing, stimulation-locked artifact suppression, EMG/EOG handling)?

4. How stable are learned spatial patterns across folds/subjects, and what happens under channel/band ablations or alternative band definitions?

5. Which components (code, trained models, tract masks, features/raw EEG) will be released at camera-ready to enable replication?

6. Would you consider softening the “foundation model” framing to match the current evidence?

---

### Official Review · Reviewer_7LvX · 2025-11-05

**Soundness:** 2
**Presentation:** 1
**Contribution:** 1
**Rating:** 2
**Confidence:** 2

**Summary:**

The paper tackles the problem of finding where to place implants for deep brain stimulation (DBS) for treatment resistant depression. Using data from previous DBS patients an SVM is trained to predict whether the correct or an incorrect part of the brain is stimulated. The learned SVM weights align with those which are biologically known to be relevant.

**Strengths:**

- This is an important, real problem.
- The paper very professionally discusses the problem and the medical/neuroscientific context for it.
- The experiments and analysis seem thorough.

**Weaknesses:**

- The paper is very hard to understand for anyone who isn’t a neuroscientist in this subfield. A much more expanded background section and simplified intro would go a very long way. There is ample space for this, the paper only ~5.5 pages long. For example, a non-expert has a hard time understanding what on and off target mean and hence what the SVM was trained on.
- Accordingly, some details which may be important are missing, e.g. in section 2.4 what were the buttons?
- I’m unsure if this paper fits ICLR, with the majority of its content being on the neuroscience/medical side and not on the machine learning side. Applied machine learning projects are of course welcome, but their main contribution should come from the application and machine learning methodology, whereas here it’s unclear whether that’s the case, especially as the application - using an SVM - is extremely simple. There is nothing bad in simplicity of course, on the contrary, but it’s unclear whether the ICLR community will have much to give or get from this paper due to its focus.
- Minor - some parts of the paper are a bit repetitive, e.g. the discussion and abstract/intro
- Minor - there are various simple formatting issues throughout, like the citations not being well formatted, having the empty Appendix, etc.

**Questions:**

NA. I’m sorry I can’t give more thorough, high quality feedback. I implore the authors to a) polish the manuscript so it’s more accessible by non-neuroscientists, more complete, and b) consider submitting to a venue which is a better fit so it may be more appropriately received.

---

### Meta-Review · Area_Chair_xXwp · 2026-01-08

**Summary:**

The reviewers overall find this paper to be under the bar for acceptance. While they all seem to agree that this is an important problem for DBS, even the most positive reviewer `gg3n` notes experimental design concerns. These alone would be disqualifying for results sections, and thereby the full paper.

As the meta-reviewer, I would like to temper these specific comments with the statement that within-subject design is possible to perform in a statistically valid way (or at least one that is community accepted), but I would also like to caution the authors that they should make it clear to the reader that they have done so, as understanding the structure of the experiments is critical for understanding the results (and again, thereby the full paper).

Beyond this, reviewer `7LvX` notes that ICLR is a methods venue, and this contribution appears to be empirical. It may be better suited in a neurology/bio-psychiatry or neuroscience conference or journal.

**Reviewer Concerns:**

Three reviewers raise experimental design concerns (all except `7LvX`, who raised the venue concern).

`PbVB` also would like to see linear baselines, which I tend to agree with.

**Reviewer Scores:**

None.

---

### Decision · Program_Chairs · 2026-01-26

Reject